# Comprehensive Analysis of Risk Factors for Periodontitis Focusing on the Saliva Microbiome and Polymorphism

**DOI:** 10.3390/ijerph18126430

**Published:** 2021-06-14

**Authors:** Naoki Toyama, Daisuke Ekuni, Daisuke Matsui, Teruhide Koyama, Masahiro Nakatochi, Yukihide Momozawa, Michiaki Kubo, Manabu Morita

**Affiliations:** 1Department of Preventive Dentistry, Okayama University Graduate School of Medicine, Dentistry and Pharmaceutical Sciences, 2-5-1 Shikata-cho, Kita-ku, Okayama 700-8558, Japan; dekuni7@md.okayama-u.ac.jp (D.E.); mmorita@md.okayama-u.ac.jp (M.M.); 2Department of Epidemiology for Community Health and Medicine, Kyoto Prefectural University of Medicine, 465 Kajii-cho, Kamigyo-ku, Kyoto 602-8566, Japan; d-matsui@koto.kpu-m.ac.jp (D.M.); tkoyama@koto.kpu-m.ac.jp (T.K.); 3Public Health Informatics Unit, Department of Integrated Health Sciences, Nagoya University Graduate School of Medicine, Nagoya 461-8673, Japan; mnakatochi@met.nagoya-u.ac.jp; 4Laboratory for Genotyping Development, RIKEN Center for Integrative Medical Sciences, 1-7-22 Suehiro-cho, Tsurumi-ku, Yokohama City 230-0045, Japan; momozawa@riken.jp (Y.M.); michiaki.kubo@riken.jp (M.K.)

**Keywords:** periodontitis, microbiota, single-nucleotide polymorphisms

## Abstract

Few studies have exhaustively assessed relationships among polymorphisms, the microbiome, and periodontitis. The objective of the present study was to assess associations simultaneously among polymorphisms, the microbiome, and periodontitis. We used propensity score matching with a 1:1 ratio to select subjects, and then 22 individuals (mean age ± standard deviation, 60.7 ± 9.9 years) were analyzed. After saliva collection, V3-4 regions of the 16S rRNA gene were sequenced to investigate microbiome composition, alpha diversity (Shannon index, Simpson index, Chao1, and abundance-based coverage estimator) and beta diversity using principal coordinate analysis (PCoA) based on weighted and unweighted UniFrac distances. A total of 51 single-nucleotide polymorphisms (SNPs) related to periodontitis were identified. The frequencies of SNPs were collected from Genome-Wide Association Study data. The PCoA of unweighted UniFrac distance showed a significant difference between periodontitis and control groups (*p* < 0.05). There were no significant differences in alpha diversity and PCoA of weighted UniFrac distance (*p* > 0.05). Two families (Lactobacillaceae and Desulfobulbaceae) and one species (*Porphyromonas gingivalis*) were observed only in the periodontitis group. No SNPs showed significant expression. These results suggest that periodontitis was related to the presence of *P. gingivalis* and the families Lactobacillaceae and Desulfobulbaceae but not SNPs.

## 1. Introduction

Periodontitis, which induces alveolar bone loss, is inflammation of periodontal tissue caused by bacterial infection. The World Health Organization reported that periodontitis is one of the main causes of tooth loss and can subsequently worsen an individual’s quality of life due to tooth loss [1]. Periodontitis affects some diseases, such as diabetes mellitus, kidney disease, premature birth, aspiration pneumonia, and arteriosclerosis [2]. Therefore, prevention of periodontitis is important to maintain both systemic and oral health.

Periodontitis is caused by a biofilm, which is a collection of any types of bacteria [3,4]. A biofilm of 1 mg includes over 1 billion bacteria. Some bacteria attach on teeth and make microcolonies. Those colonies gather and make biofilms. In biofilms, bacteria interact and obtain some benefits, including a broader habitat range, efficient metabolism, increased tolerance, and enhanced virulence [5]. The concept of microbiome analysis in periodontal science is to investigate the diversity of communities of bacteria.

It is important to control risk factors for periodontitis to prevent disease. Previous studies reported several risk factors, including smoking, diabetes mellitus, stress, and obesity [2]. Recently, studies reported the microbiome [6] and polymorphisms [7] as risk factors for periodontitis. Polymorphisms are the most common types of genetic variation among individuals [8]. They occur once in 500–1000 nucleotides. Most polymorphisms have no effect on periodontal health, but some polymorphisms are associated with risks of disease. Munz et al. showed that *MTND1P5* (re16870060-G) and *SHISA9* (rs729876-T) were associated with the risk of periodontitis [7].

Dentists have been struggling to prevent worsening periodontitis. Because the mechanism of worsening periodontitis is a host–parasite interaction, we have to pay attention to both the oral microbiome and host factors. However, few studies showed associations between the microbiome (parasite) and polymorphisms (host) at the same time. We hypothesized that there are associations between polymorphisms, the microbiome, and periodontitis. The objective of the present study was to assess associations simultaneously between polymorphisms, the microbiome, and periodontitis.

## 2. Materials and Methods

### 2.1. Study Population

The present study was a cross-sectional study including individuals who were enrolled in the Japan Multi-Institutional Collaborative Cohort Study (J-MICC study) second survey in the Kyoto area from 2013 to 2017. Participants underwent oral examinations, were measured for height and weight, provided saliva and blood samples, and answered self-reported questionnaires. The inclusion criterion was participants who provided saliva samples. The exclusion criteria were participants who had no teeth and could not be evaluated for the community periodontal index (CPI) and who had CPI = 1–3.

### 2.2. Ethical Procedures and Informed Consent

The study was approved by the Institutional Ethics Committee of Kyoto Prefectural University of Medicine (approval number: RBMR-E-36-8, in 2013) and was conducted in accordance with the principles of the Declaration of Helsinki. All participants provided written, informed consent before participation.

### 2.3. Oral Examinations

One general dentist checked dental status using a dental mirror and explorer under artificial light. Periodontal status was assessed using the CPI (World Health Organization, 4th edition). The CPI was measured for 10 teeth (maxilla: right first and second molars, right central incisor, left first and second molars; mandible: right first and second molars, left central incisor, left first and second molars). A CPI of 0 was defined as no periodontitis (control group), and a CPI of 4 was defined as periodontitis (periodontitis group).

### 2.4. Questionnaires

A self-administered questionnaire was used to determine whether participants had risk factors for periodontitis. Based on the Japanese Association of Periodontology Clinical Practice Guideline for the Periodontal Treatment, 2015 [2], we checked for risk factors such as age, sex, stress, smoking, and diabetes mellitus as follows:Have you felt stress during the past year? (Severe/Moderate/Mild/None)Are you a smoker? (Yes/Past/No)Do you take at least one medication per week to lower blood sugar levels? (Yes/No)

We combined “Severe” and “Moderate” responses as positive awareness, and “Mild” and “None” responses as negative awareness to evaluate stress. Participants who took medicines to lower blood sugar levels were defined as having diabetes mellitus.

### 2.5. Salivary Microbiome Analysis

To investigate microbiome composition, saliva samples were collected. Saliva was collected by passive drool through a 1 inch straw into a vial. Approximately 1.0 mL of saliva was collected into a 10 mL centrifuge tube (Salivette^®^, Sarstedt AG & Co., Nümbrecht, Germany) from all examinees. Saliva samples in centrifuge tubes were stored at −20 °C until assayed. To prevent contamination, sterilized tubes were used for saliva collection.

DNA sequencing was performed based on the 16S metagenomics sequencing library preparation protocol [9] at the Oral Microbiome Center (Taniguchi Dental Clinic, Kagawa, Japan). First, DNA was extracted from saliva. Saliva was suspended using lysis buffer and crushed for 3 min. After centrifugation, DNA was extracted using GenFind 2.0 (Beckman Coulter Inc., Brea, CA, USA). During the first PCR, the V3 and V4 regions of the 16S rRNA gene were amplified using primers 341F (forward primer: 5′-TCGTCGGCAGCGTCAGATGTGTATAAGAGACAGNNNCCTACGGGNGGCWGCAG-3′) and 806R (reverse primer: 5′-GTCTCGTGGGCTCGGAGATGTGTATAAGAGACAGNNNGACTACHVGGGTATCTAATCC-3′), using Kapa HiFi HotStart 2× ReadyMix DNA polymerase (Kapa Biosystems Ltd., London, UK). Cycle conditions were 95 °C (3 min), 28 cycles of 95 °C (30 s), 55 °C (30 s), and 72 °C (30 s), followed by a final extension of 72 °C (5 min). After extension, primers were removed using 50 μL AMPure XP. Thereafter, the second PCR was performed using primers including the 8 nt identifying index. Cycle conditions were 95 °C (3 min), 8 cycles of 95 °C (30 s), 55 °C (30 s), and 72 °C (30 s), followed by a final extension of 72 °C (5 min). Next-generation sequencing was performed using the obtained sequences and the MiSeq platform (MiSeq Reagent V3 600 cycles, Illumina, San Diego, CA, USA). Obtained reads showed that the mean Phred quality score 30 was 81.4%. Low-quality reads were discarded. After sequencing, UPARSE [10] was used to generate operational taxonomic units (OTUs), UCLUST [11] to divide the sequence into a cluster, and BLAST to search for homology (reference database, Greengenes 13.5).

### 2.6. Genotyping

The 14,539 study participants from the 12 areas of the J-MICC study were genotyped at the RIKEN Center for Integrative Medical Sciences using the Human OmniExpressExome-8 v1.2 BeadChip array (San Diego, CA, USA) [12]. The quality control of samples and SNPs was performed as in a previous study [8]. Briefly, data were subjected to quality control procedures, by which SNPs with a call rate of <0.98 or a Hardy–Weinberg equilibrium *p*-value <1 × 10^−6^ or a low minor allele frequency <0.01 were filtered out. Genotype imputation was performed using SHAPIT [13] and Minimac3 [14] software based on the 1000 Genomes Project cosmopolitan reference panel (phase 3). After genotype imputation, variants with an R^2^ < 0.3 were excluded, resulting in 12,617,547 variants. Of the SNPs that passed quality control, 51 SNPs related to periodontitis were identified [15,16,17,18,19,20,21,22].

### 2.7. Assessment of Other Factors

Data were collected from participants’ medical records for triglyceride, blood glucose, and HbA1c levels, and the body mass index (BMI) was calculated as weight (kg) divided by the square of height (m^2^).

### 2.8. Statistical Analysis

Participants were assigned to the periodontitis group or control group using propensity score matching with a 1:1 ratio [23] to adjust for confounders of periodontitis. Propensity scores were calculated using logistic regression models adjusted for age, sex, smoking status, stress, diabetes mellitus, and BMI, which were confounders. To check balanced covariates between the periodontitis and control groups, Fisher’s exact test and two-sample *t*-tests were performed after propensity score matching. Then, the variance inflation factor (VIF) was used to assess the multicollinearity of parameters used for propensity score matching. No multicollinearity of parameters was defined as VIF < 10. Model fitness using propensity score matching was assessed by the Hosmer–Lemeshow test.

Alpha (richness and evenness of bacterial taxa within a community) and beta (ecological distances between samples) diversities were analyzed to show the difference in microbiome composition between the periodontitis and control groups. Alpha diversities were assessed by the Shannon index, Simpson’s index, Chao1, and the abundance-based coverage estimator (ACE) [24]. Beta diversities were assessed by principal coordinate analysis (PCoA) based on weighted and unweighted UniFrac distances [25]. The weighted UniFrac distance reflects the difference in abundant lineages. Unweighted UniFrac distances show the difference in rare lineages [26]. The differences in microbial composition between the two groups were assessed by Adonis and ANOSIM, and the difference in dispersion was judged by Permdisp2 using “Calypso” (http://cgenome.net:8080/calypso-8.84/faces/uploadFiles.xhtml/; accessed on 6 March 2020).

Linear discriminant analysis effect size (LEfSe) showed differences in OTU richness between periodontitis and control groups using “Galaxy” (https://huttenhower.sph.harvard.edu/galaxy/; accessed on 6 March 2020). A cutoff value of 2.0 was used. For LDA scores >2.0, Mann–Whitney U tests were used to determine significant differences between the two groups. The difference in SNP expression between two groups was assessed using Fisher’s exact test.

Statistical analyses were conducted using SPSS version 22 (IBM, Tokyo, Japan) for Fisher’s exact test, the chi-squared test, the two-sample *t-*test, and the Mann–Whitney U test. All *p*-values <0.05 were considered significant. To adjust for multiple comparisons, the *q*-value was calculated using the Benjamini and Hochberg false discovery rate [27] based on the results of the *p*-value. All *q*-values <0.05 were considered significant.

### 2.9. Data Availability

Requests to access the dataset should be addressed to the J-MICC Study Central Office via the corresponding author of this paper.

## 3. Results

### 3.1. Participants’ Characteristics

Figure 1 shows the study flowchart. Overall, 385 of 3917 participants provided saliva samples in the J-MICC study. Then, 31 participants who met the inclusion criteria and with CPI = 0 or 4 were selected, and 22 participants (11 males and 11 females; mean age ± standard deviation (SD), 60.7 ± 9.9 years) were finally included after propensity score matching. No significant differences in risk factors for periodontitis (age, sex, smoking status, stress, diabetes mellitus, and BMI) or blood test items were observed between the periodontitis and control groups (Table 1). All parameters showed VIF < 10 (data not shown). The model showed good fit (Hosmer–Lemeshow test; *p* = 0.554).

### 3.2. Salivary Microbiome Analysis

A total of 2,314,918 reads were obtained from 22 saliva samples, and 1,993,572 quality-filtering reads (mean ± SD: 90,617 ± 19,112) were used for analysis. A total of 471 OTUs were obtained based on 97% sequence similarity, and 12 phyla, 23 classes, 36 orders, 58 families, 112 genera, and 349 species were identified. In alpha diversity, there were no indices that showed significance (Shannon index, *p* = 0.59; Simpson index, *p* = 0.24; Chao1, *p* = 0.18; ACE, *p* = 0.18) (Appendix A).

Figure 2 shows the results of the PCoA of weighted and unweighted UniFrac distances. For unweighted UniFrac distances, there were significant differences between the periodontitis and control groups (ANOSIM, *p* = 0.004, R = 0.18; Adonis, *p* = 0.006, R^2^ = 0.105). However, there was no significant difference in the homogeneity of dispersions between the two groups (Permdisp2, *p* = 0.589). In contrast, for weighted UniFrac distances, there were no significant differences (ANOSIM, *p* = 0.136, R = 0.066; Adonis, *p* = 0.347, R^2^ = 0.049).

Figure 3 shows LEfSe at the species level. LDA scores >2.0 were observed for 1 phylum, 5 classes, 6 orders, 10 families, 14 genera, and 26 species in the periodontitis group, and 2 genera and 10 species in the control group. Significant differences were observed in one phylum (Synergistetes), five classes (Coriobacteriia, Bacteroidetes [C-1], Mollicutes, Deltaproteobacteria, Synergistia), five orders (Coriobacteriales, Bacteroidetes [O-1], Mycoplasmatales, Desulfobacterales, Synergistales), seven families (Atopobiaceae, Bacteroidetes [F-1], Lactobacillaceae, Mycoplasmataceae, Desulfobulbaceae, TM7 [F-2], Synergistaceae), two genera (*Olsenella*, *Fretibacterium*), and one species (*Porphyromonas gingivalis*) (Table 2).

### 3.3. Genotyping

No SNPs showed significantly different expressions between the two groups (*q* > 0.05, data not shown). Associations between periodontitis and SNPs are shown in Appendix A.

## 4. Discussion

The microbiome and polymorphisms for periodontitis were investigated in 22 individuals. Two families and one species were specifically confirmed in a periodontitis group. However, single-nucleotide polymorphisms were not related to periodontitis. To the best of our knowledge, the present study is the first to assess the associations among periodontitis, the microbiome, and genotyping. Beta diversity based on unweighted UniFrac distance showed a significant difference between the periodontitis and control groups, but the weighted UniFrac distance did not. Alpha diversity and SNPs were not associated with periodontitis. Unweighted UniFrac distance considers only species presence and is most efficient in detecting abundance change in rare lineages [26]. The present study showed that two families (Lactobacillaceae and Desulfobulbaceae) and one species (*P. gingivalis*) existed only in the periodontitis group (Table 2). Thus, the presence of the families Lactobacillaceae and Desulfobulbaceae and of *P. gingivalis* could be related to periodontitis. These results also suggested that the microbiome affects periodontitis more than SNPs do clinically.

*P. gingivalis* is the most abundant species in the subgingival microbiota of patients with periodontitis and is related to chronic periodontitis [28]. Damgaard et al. reported that the presence of *P. gingivalis* in saliva is significantly associated with periodontitis [28]. Guerra et al. reported that *P. gingivalis* in the salivary microbiome is a risk factor for periodontitis [6]. The present results support the existence of *P. gingivalis* as a risk factor for periodontitis.

The families Lactobacillaceae and Desulfobulbaceae were significantly more frequent in the periodontitis group than in the control group. Lactobacillaceae is isolated in the mouth, but its contribution to oral health has not been determined [29]. Desulfobulbaceae is a sulfate-reducing bacterium that is isolated from periodontal pockets [30]. Because few studies showed periodontitis associated with the families Lactobacillaceae and Desulfobulbaceae, further investigations are needed to clarify the associations.

Beta diversity in unweighted UniFrac distance was found to be significantly different between the periodontitis and control groups, whereas alpha and beta diversities in weighted UniFrac distance were not. Schulz et al. reported a significant difference in beta diversity, but not in alpha diversity, for periodontitis [31]. Acharya et al. also showed that alpha diversity was not significantly different between periodontitis and healthy gingiva [32]. The present findings support these findings. In contrast, Takeshita et al. showed that alpha diversity was significantly different between periodontitis and healthy gingiva [33]. One potential explanation for discrepancies in the results for diversity might be the effect of the circadian rhythm. Takayasu et al. reported that genera accounting for 79.3% of the oral microbiome change roughly every 24 h [34]. The circadian rhythm resulted in a greater number of bacteria at the phylum level (Firmicutes and Bacteroidetes) [34]. Therefore, the circadian rhythm might affect alpha and beta diversities in weighted UniFrac distance, which is based on the number of bacteria.

The results of the present study might have clinical relevance. The prevalence of periodontitis was significantly associated with the salivary microbiome, but not with SNPs. Some studies showed the association between SNPs and periodontitis [15,16,17,18,19,20,21,22], but few studies assessed the relationships among polymorphisms, the microbiome, and periodontitis simultaneously. The results showed that the microbiome, rather than SNPs, affected the prevalence of periodontitis. The effect of individual SNPs on periodontitis might be small. A previous study showed that many SNPs related to periodontitis had small odds ratios (<1.5) [35]. Therefore, clinicians should pay attention to microbiome composition to prevent periodontitis.

The present study had some strengths. First, associations among periodontitis, the microbiome, and SNPs were examined simultaneously. Second, covariates were reduced because propensity score matching was used [23]. Since periodontitis is a multifactorial disease, propensity score matching is useful to investigate the associations among periodontitis, the microbiome, and SNPs.

Some limitations must be considered when interpreting the results. First, the conditions of saliva collection were not controlled. The oral microbiome is affected by diurnal rhythm, oral health behavior, and food intake [36]. For generalization, it might be important to control the timing of saliva collection, food consumption, and oral health behaviors. The interval between saliva collection and eating was checked immediately before saliva collection, and it was not significantly different between the two groups (periodontitis group, 264.6 ± 79.8 min; control group, 241.2 ± 72.7 min; two-sample *t-*test, *p* = 0.585). However, when participants brushed their teeth before saliva collection was not known. Second, the causal association is unclear because the present study was a cross-sectional study. Therefore, it is unclear whether the present results are related to the onset or the progression of periodontitis. Third, there might be sampling bias. Saliva samples were provided from only a portion of participants in the J-MICC study (9.8%). Therefore, caution is needed before generalizing the present findings.

## 5. Conclusions

In conclusion, periodontitis was related to the presence of *P. gingivalis* and the families Lactobacillaceae and Desulfobulbaceae, but not to SNPs. The present results suggest that the oral microbiome, not polymorphism, is a risk factor for periodontitis, and that clinicians should pay more attention to microbiome composition than to host factors in the routine work of periodontal examination and diagnosis. Further investigations are needed to clarify the role of the families Lactobacillaceae and Desulfobulbaceae in periodontitis.

## Figures and Tables

**Figure 1 ijerph-18-06430-f001:**
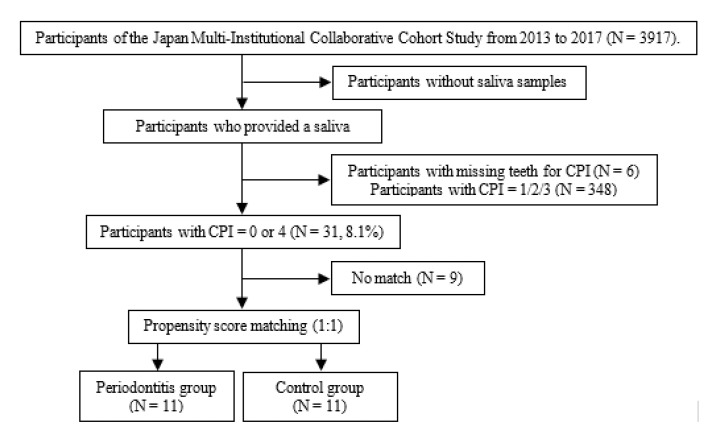
Study flowchart. CPI, community periodontal index.

**Figure 2 ijerph-18-06430-f002:**
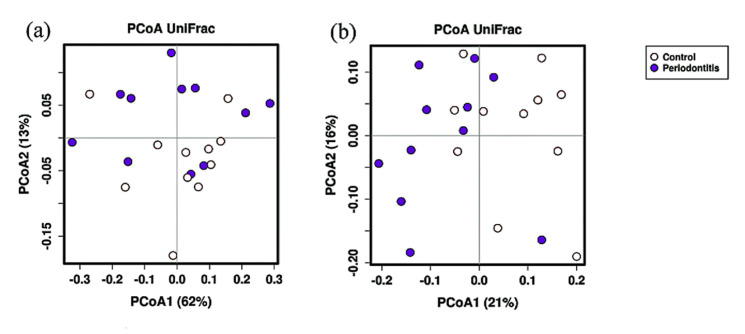
Principal coordinate analysis (PCoA) to assess the difference in beta diversity between the periodontitis and control groups. Plots were based on weighted UniFrac distance (**a**), unweighted UniFrac distance (**b**), and relative abundances of bacterial operational taxonomic units.

**Figure 3 ijerph-18-06430-f003:**
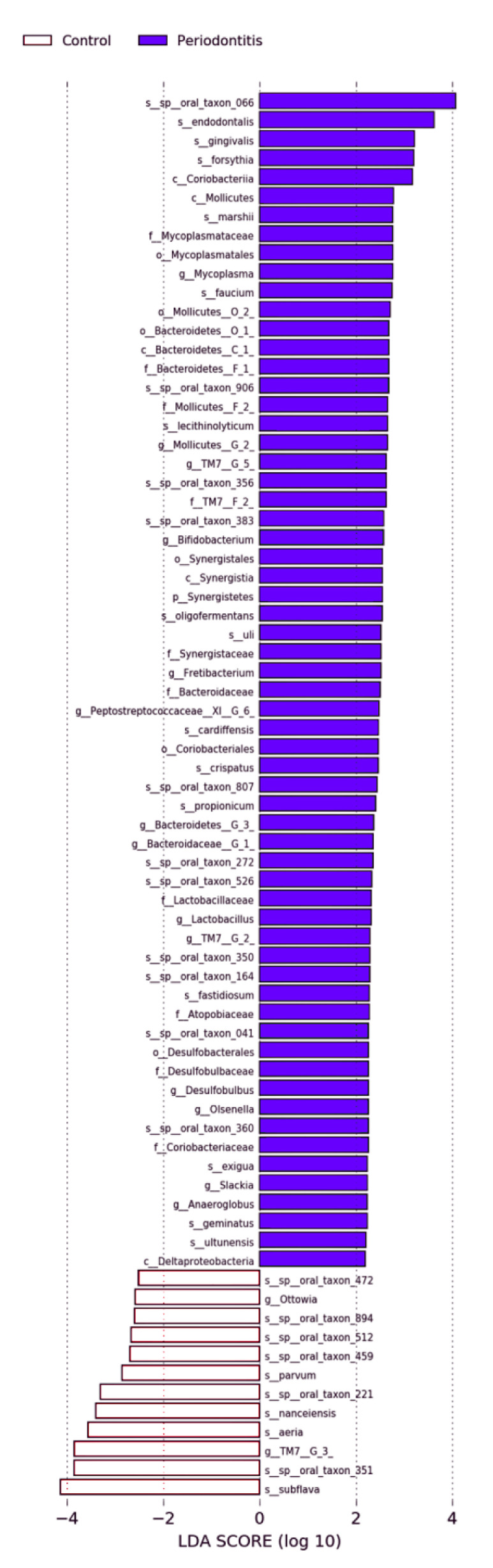
Graphics of linear discriminant analysis (LDA) effect size (LEfSe) for the periodontitis and control groups. Horizontal bars show the effect size for each taxon. The length of the bar indicates the log10 transformed LDA score, indicated by vertical dotted lines. The threshold on the logarithmic LDA score for discriminative features was set to 2.0. The taxon of bacteria with significant change in relative abundance (*n* = 74, linear discriminant analysis, *p* < 0.05) is written alongside the horizontal lines. The name of the taxon level was abbreviated as follows: p, phylum; c, class; o, order; f, family; g, genus; and s, species.

**Table 1 ijerph-18-06430-t001:** Differences in parameters between the periodontal disease and control groups.

		Periodontal Disease Group	Control Group	*p*
		N = 11	N = 11	
Gender	Male	6 (54.5) ^a^	5 (45.5)	1.000 ^b^
	Female	5 (45.5)	6 (54.5)	
Passive stress	Negative	5 (45.5)	2 (18.2)	0.361
	Positive	6 (54.5)	9 (81.8)	
Smoking status	No	3 (27.3)	7 (63.6)	0.061 ^c^
	Past	4 (36.4)	4 (36.4)	
	Current	4 (36.4)	0 (0)	
Diabetes	No	9 (81.8)	11 (100)	0.333
	Past	1 (9.1)	0 (0)	
	Current	1 (9.1)	0 (0)	
Age		64.3 ± 7.6 ^d^	57.1 ± 11.0	0.091 ^e^
Body mass index		23.9 ± 3.3	21.3 ± 2.7	0.051
Triglyceride (mg/dL)		102.5 ± 34.7	89.2 ± 48.8	0.470
HbA1c (%)		5.8 ± 0.3	5.0 ± 0.3	0.065
Blood glucose (mg/dL)		100.8 ± 19.3	92.7 ± 13.0	0.263

^a^ N (%), ^b^ Fisher’s exact test, ^c^ chi-square test, ^d^ mean ± SD, ^e^ two sample *t* test. HbA1c, hemoglobin A1c.

**Table 2 ijerph-18-06430-t002:** Differences in operational taxonomic units (OTUs) between the periodontitis and control groups.

OTU	Periodontitis			Control			*p* ^a^	*q* ^b^
Species	Median	25th Percentiles	75th Percentiles	Median	25th Percentiles	75th Percentiles		
*Actinomyces cardiffensis*	1.31 × 10^−4^	0	2.48 × 10^−4^	0	0	0	0.047	0.074
*Rothia aeria*	1.20 × 10^−3^	9.45 × 10^−5^	1.15 × 10^−2^	8.36 × 10^−3^	2.28 × 10^−3^	2.16 × 10^−2^	0.047	0.071
*Propionibacterium propionicum*	0	0	2.74 × 10^−5^	4.58 × 10^−5^	2.51 × 10^−5^	8.32 × 10^−5^	0.023	0.075
*Olsenella* sp. *oral taxon 807*	4.87 × 10^−5^	0	1.29 × 10^−4^	0	0	0	0.065	0.087
*Olsenella uli*	5.47 × 10^−5^	0	1.31 × 10^−4^	0	0	2.13 × 10^−5^	0.040	0.076
*Slackia exigua*	2.25 × 10^−4^	2.75 × 10^−5^	6.58 × 10^−4^	3.02 × 10^−5^	0	5.12 × 10^−5^	0.040	0.072
*Bacteroidaceae* [G-1] sp. *oral taxon 272*	3.56 × 10^−4^	2.19 × 10^−5^	7.54 × 10^−4^	2.48 × 10^−5^	0	1.07 × 10^−4^	0.047	0.068
*Porphyromonas endodontalis*	7.06 × 10^−3^	1.14 × 10^−3^	2.50 × 10^−2^	1.12 × 10^−3^	0	3.69 × 10^−3^	0.034	0.068
*Porphyromonas gingivalis*	6.61 × 10^−4^	4.96 × 10^−4^	2.79 × 10^−3^	0	0	0	<0.001	0.007
*Tannerella forsythia*	1.73 × 10^−3^	1.05 × 10^−3^	8.62 × 10^−3^	4.29 × 10^−4^	4.22 × 10^−5^	1.29 × 10^−3^	0.028	0.072
*Prevotella marshii*	2.36 × 10^−5^	0	8.26 × 10^−5^	0	0	0	0.028	0.067
*Prevotella nanceiensis*	8.27 × 10^−4^	1.26 × 10^−4^	6.66 × 10^−3^	8.12 × 10^−3^	2.73 × 10^−3^	1.60 × 10^−2^	0.023	0.069
*Prevotella* sp. *oral taxon 472*	0	0	2.75 × 10^−5^	1.02 × 10^−4^	6.45 × 10^−5^	1.49 × 10^−4^	0.007	0.084
*Prevotella* sp. *oral taxon 526*	1.31 × 10^−4^	0	6.76 × 10^−4^	0	0	0	0.019	0.086
*Lactobacillus crispatus*	0	0	1.08 × 10^−4^	0	0	0	0.151	0.160
*Lactobacillus ultunensis*	5.90 × 10^−5^	0	1.44 × 10^−4^	0	0	0	0.065	0.084
*Streptococcus oligofermentans*	8.05 × 10^−5^	2.53 × 10^−5^	9.74 × 10^−5^	2.13 × 10^−5^	0	2.77 × 10^−5^	0.005	0.090
*Streptococcus* sp. *oral taxon 066*	3.28 × 10^−2^	1.34 × 10^−2^	5.78 × 10^−2^	1.28 × 10^−2^	9.79 × 10^−3^	2.09 × 10^−2^	0.013	0.078
*Catonella* sp. *oral taxon 164*	2.74 × 10^−5^	0	1.31 × 10^−4^	0	0	0	0.076	0.094
*Oribacterium parvum*	0	0	0	2.29 × 10^−5^	0	4.30 × 10^−5^	0.101	0.121
*Peptostreptococcaceae* [XI][G-1] sp. *oral taxon 383*	0	0	7.18 × 10^−5^	0	0	0	0.151	0.151
*Mollicutes* [G-2] sp. *oral taxon 906*	1.61 × 10^−5^	0	1.10 × 10^−4^	0	0	0	0.101	0.117
*Mycoplasma faucium*	6.68 × 10^−4^	0	1.26 × 10^−3^	0	0	2.15 × 10^−5^	0.013	0.067
*Anaeroglobus geminatus*	3.59 × 10^−5^	0	3.14 × 10^−4^	0	0	0	0.019	0.076
*Leptotrichia* sp. *oral taxon 221*	4.87 × 10^−5^	0	7.67 × 10^−4^	7.46 × 10^−4^	1.37 × 10^−4^	5.99 × 10^−3^	0.040	0.069
*Ottowia* sp. *oral taxon 894*	0	0	2.74 × 10^−5^	1.07 × 10^−4^	2.56 × 10^−5^	1.60 × 10^−4^	0.047	0.065
*Kingella* sp. *oral taxon 459*	0	0	0	0	0	1.51 × 10^−4^	0.151	0.151
*Neisseria subflava*	1.89 × 10^−4^	1.61 × 10^−5^	3.02 × 10^−3^	5.72 × 10^−2^	2.60 × 10^−2^	7.43 × 10^−2^	0.008	0.058
*Desulfobulbus* sp. *oral taxon 041*	5.51 × 10^−5^	0	2.51 × 10^−4^	0	0	0	0.028	0.063
*Aggregatibacter* sp. *oral taxon 512*	2.36 × 10^−5^	0	1.45 × 10^−4^	1.54 × 10^−4^	2.51 × 10^−5^	1.26 × 10^−3^	0.040	0.065
*TM7* [G-2] sp. *oral taxon 350*	0	0	2.20 × 10^−4^	0	0	0	0.133	0.150
*TM7* [G-3] sp. *oral taxon 351*	1.04 × 10^−2^	2.38 × 10^−3^	1.90 × 10^−2^	2.18 × 10^−2^	1.36 × 10^−2^	3.79 × 10^−2^	0.023	0.064
*TM7* [G-5] sp. *oral taxon 356*	3.58 × 10^−4^	1.77 × 10^−4^	1.13 × 10^−3^	1.26 × 10^−4^	0	3.05 × 10^−4^	0.019	0.068
*Treponema lecithinolyticum*	0	0	1.09 × 10^−4^	0	0	0	0.133	0.145
*Fretibacterium fastidiosum*	8.75 × 10^−5^	4.87 × 10^−5^	1.89 × 10^−4^	1.79 × 10^−5^	0	6.04 × 10^−5^	0.028	0.059
*Fretibacterium* sp. *oral taxon 360*	2.92 × 10^−4^	5.90 × 10^−5^	7.66 × 10^−4^	2.56 × 10^−5^	0	9.16 × 10^−5^	0.007	0.063
**Genus**								
*Bifidobacterium*	0	0	3.03 × 10^−4^	0	0	0	0.133	0.142
*Olsenella*	1.75 × 10^−4^	2.75 × 10^−5^	3.29 × 10^−4^	0	0	3.57 × 10^−5^	0.002	0.032
*Slackia*	2.25 × 10^−4^	2.75 × 10^−5^	6.58 × 10^−4^	3.02 × 10^−5^	0	5.12 × 10^−5^	0.040	0.058
*Bacteroidetes* [G-3]	2.87 × 10^−4^	4.38 × 10^−5^	1.02 × 10^−3^	0	0	1.83 × 10^−4^	0.034	0.060
*Bacteroidaceae* [G-3]	3.56 × 10^−4^	2.19 × 10^−5^	7.54 × 10^−4^	2.48 × 10^−5^	0	1.07 × 10^−4^	0.047	0.063
*Lactobacillus*	6.56 × 10^−5^	0	2.51 × 10^−4^	0	0	0	0.034	0.054
*Peptostreptococcaceae* [XI][G-6]	1.64 × 10^−4^	1.09 × 10^−4^	1.47 × 10^−3^	4.26 × 10^−5^	0	1.25 × 10^−4^	0.010	0.053
*Mollicutes* [G-2]	1.61 × 10^−5^	0	1.10 × 10^−4^	0	0	0	0.101	0.115
*Mycoplasma*	1.05 × 10^−3^	1.31 × 10^−4^	1.51 × 10^−3^	1.24 × 10^−4^	2.77 × 10^−5^	5.63 × 10^−4^	0.023	0.061
*Anaeroglobus*	3.59 × 10^−5^	0	3.14 × 10^−4^	0	0	0	0.019	0.076
*Ottowia*	0	0	2.74 × 10^−5^	1.07 × 10^−4^	2.56 × 10^−5^	1.60 × 10^−4^	0.047	0.058
*Desulfobulbus*	5.51 × 10^−5^	0	2.51 × 10^−4^	0	0	0	0.028	0.056
*TM7* [G-2]	0	0	2.20 × 10^−4^	0	0	0	0.133	0.133
*TM7* [G-3]	1.04 × 10^−2^	2.38 × 10^−3^	1.90 × 10^−2^	2.18 × 10^−2^	1.36 × 10^−2^	3.79 × 10^−2^	0.023	0.053
*TM7* [G-5]	3.58 × 10^−4^	1.77 × 10^−4^	1.13 × 10^−3^	1.26 × 10^−4^	0	3.05 × 10^−4^	0.019	0.061
*Fretibacterium*	4.14 × 10^−4^	1.97 × 10^−4^	1.65 × 10^−3^	5.02 × 10^−5^	1.79 × 10^−5^	1.81 × 10^−4^	0.004	0.032
**Family**								
Atopobiaceae	1.75 × 10^−4^	2.75 × 10^−5^	3.29 × 10^−4^	0	0	3.57 × 10^−5^	0.002	0.020
Coriobacteriaceae	2.46 × 10^−4^	2.75 × 10^−5^	6.58 × 10^−4^	3.02 × 10^−5^	1.79 × 10^−5^	5.12 × 10^−5^	0.047	0.052
Bacteroidetes [F-1]	8.77 × 10^−4^	1.61 × 10^−4^	2.55 × 10^−3^	1.07 × 10^−4^	0	2.77 × 10^−4^	0.034	0.049
Bacteroidaceae	7.09 × 10^−4^	1.38 × 10^−4^	1.21 × 10^−3^	9.14 × 10^−5^	0	1.83 × 10^−4^	0.019	0.063
Lactobacillaceae	6.56 × 10^−5^	0	2.51 × 10^−4^	0	0	0	0.034	0.043
Mollicutes [F-1]	1.61 × 10^−5^	0	1.10 × 10^−4^	0	0	0	0.101	0.101
Mycoplasmataceae	1.05 × 10^−3^	1.31 × 10^−4^	1.51 × 10^−3^	1.24 × 10^−4^	2.77 × 10^−5^	5.63 × 10^−4^	0.023	0.046
Desulfobulbaceae	5.51 × 10^−5^	0	2.51 × 10^−4^	0	0	0	0.028	0.047
TM7 [F-2]	3.58 × 10^−4^	1.77 × 10^−4^	1.13 × 10^−3^	1.26 × 10^−4^	0	3.05 × 10^−4^	0.019	0.048
Synergistaceae	4.14 × 10^−4^	1.97 × 10^−4^	1.65 × 10^−3^	5.02 × 10^−5^	1.79 × 10^−5^	1.81 × 10^−4^	0.004	0.020
**Order**								
Coriobacteriales	4.18 × 10^−4^	1.65 × 10^−4^	7.51 × 10^−4^	5.02 × 10^−5^	2.15 × 10^−5^	7.45 × 10^−5^	0.005	0.015
Bacteroidetes [O-1]	8.77 × 10^−4^	1.61 × 10^−4^	2.55 × 10^−3^	1.07 × 10^−4^	0	2.77 × 10^−4^	0.034	0.041
Mollicutes [O-2]	1.61 × 10^−5^	0	1.10 × 10^−4^	0	0	0	0.101	0.101
Mycoplasmatales	1.05 × 10^−3^	1.31 × 10^−4^	1.51 × 10^−3^	1.24 × 10^−4^	2.77 × 10^−5^	5.63 × 10^−4^	0.023	0.046
Desulfobacterales	5.51 × 10^−5^	0	2.51 × 10^−4^	0	0	0	0.028	0.042
Synergistales	4.14 × 10^−4^	1.97 × 10^−4^	2.23 × 10^−3^	5.02 × 10^−5^	1.79 × 10^−5^	1.81 × 10^−4^	0.002	0.012
**Class**								
Coriobacteriia	9.98 × 10^−4^	1.97 × 10^−4^	5.06 × 10^−3^	5.02 × 10^−5^	0	5.02 × 10^−5^	0.005	0.013
Bacteroidetes [C-1]	8.77 × 10^−4^	1.61 × 10^−4^	2.55 × 10^−3^	3.02 × 10^−5^	0	1.81 × 10^−4^	0.034	0.034
Mollicutes	1.17 × 10^−3^	1.65 × 10^−4^	1.51 × 10^−3^	4.96 × 10^−5^	0	1.74 × 10^−4^	0.016	0.020
Deltaproteobacteria	8.26 × 10^−5^	0	2.51 × 10^−4^	5.54 × 10^−4^	2.13 × 10^−5^	1.13 × 10^−3^	0.007	0.012
Synergistia	4.14 × 10^−4^	1.97 × 10^−4^	2.23 × 10^−3^	9.14 × 10^−4^	3.05 × 10^−5^	9.90 × 10^−4^	0.002	0.010
**Phylum**								
Synergistetes	4.14 × 10^−4^	1.97 × 10^−4^	2.23 × 10^−3^	5.02 × 10^−5^	1.79 × 10^−5^	1.81 × 10^−4^	0.002	

^a^ Mann–Whitney *U* test, ^b^ adjusted p-values by Benjamini and Hochberg’s false discovery rate (FDR).

## Data Availability

Requests to access the dataset should be addressed to the J-MICC Study Central Office via the corresponding author of this paper.

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
