# Peer review of "Comprehensive Analysis of Risk Factors for Periodontitis Focusing on the Saliva Microbiome and Polymorphism"

_ijerph, 2021, doi:10.3390/ijerph18126430_

Round 1
Reviewer 1 Report
Introduction: You state that “periodontitis” is “inflammation of tooth tissue.” This implies that the hard dental tissues (enamel, dentin) are inflamed; the periodontal tissues should be referred to instead.
You state that prior studies have “reported microbiome [3] and polymorphism [4] as risk factors of periodontitis. However, few studies have exhaustively assessed relationships between polymorphism, microbiome, and periodontitis.” Instead of using the term “exhaustively,” I would suggest that you specify the research gap that your study addresses.
Results: You state the following: “Overall, 385 of 3,917 participants provided saliva 162 samples in the J-MICC study. Then, we selected 31 participants who met inclusion criteria and finally included 22 participants.” However, in the Methods section you state that the “inclusion criterion was participants who provided saliva samples.” As 385 participants provided saliva samples, please clarify on what basis the sample size was reduced to 31, and then 22 participants.
Numerous grammatical errors are evident throughout the manuscript. I would suggest that this manuscript be professionally edited for language.
Reviewer 2 Report
Dear Authors,
congratulations for the study. However the presentation should be implemented to make the article more appealing.
Indeed the intro is quite scarce and need to be enriched so that the people who are not familiar with the concept of polimorphism and periodontal sciences can be better informed also to understand Your research.
Indeed, the whole concept of oral biofilm is missing
https://pubmed.ncbi.nlm.nih.gov/26416306/
https://pubmed.ncbi.nlm.nih.gov/31725203/
https://pubmed.ncbi.nlm.nih.gov/28805207/
Results:
an introductive paragraph summarizing and semplyifing Your results would help to better understand them
Conclusion is really really really scarce.
Your results are good and whorty to be read. Improve and enrich Your conclusions and Your take home message
Round 2
Reviewer 1 Report
The manuscript has been improved after modifications. However, additional English language editing would be beneficial.
Author Response
Thank you for your comments.
Our manuscript was edited by a native English speaker at June 8,2021.
I attach a certificate of native check.
